# Silencing NRF2 enhances arsenic trioxide-induced ferroptosis in hepatocellular carcinoma cells

Mi Huang[1,2☉], Duanzhuo Li[1,2☉], Zhengzhen Xia[2,3☉], Shengjie Liao[1], Wenxia Si[1], Chao Yuan[1], Yanli Liao[1], Weibin Wu[1], Minshu Jiang[3], Xin Yu[1]*, Yi Quan[2,3]*

1 Department of Scientific Research and Experiment Center, Zhaoqing Medical College, Guangdong, People's Republic of China, 2 Department of Oncology, The First People's Hospital of Zhaoqing Affiliated to Zhaoqing Medical College, Guangdong, People's Republic of China, 3 The First Clinical Medical School, Guangdong Medical University, Zhanjiang, Guangdong, People's Republic of China

☉ These authors contributed equally to this work.
* quany_i@sina.com (YQ); yuxin@zqmc.edu.cn (XY)

## Abstract

### Objective

Hepatocellular carcinoma (HCC) is a leading cause of cancer-related deaths worldwide, with high mortality rates partially due to limited therapeutic options and drug resistance. Arsenic trioxide (ATO), a compound clinically proven for acute promyelocytic leukemia (APL), has garnered attention for its emerging efficacy in solid tumors, including HCC. However, the molecular mechanisms driving ATO's antitumor activity in HCC remain incompletely understood. In this study, we aimed to elucidate the ferroptosis-dependent effects of ATO on HCC and and propose a potential therapeutic strategy.

### Methods

The response of HCC cells to ATO was evaluated using cell viability, wound healing, colony formation, Transwell migration assays, and cell cycle analysis. ATO-induced ferroptosis was assessed by measuring lipid peroxidation (via C11-BODIPY staining), intracellular iron levels, and malondialdehyde (MDA) production. Western blotting was performed to quantify protein levels of NRF2, HO-1, SLC7A11, and GPX4; immunofluorescence staining was employed to determine NRF2 subcellular localization.

### Results

ATO exhibited significant cytotoxicity and inhibited the progression of HCC cells. Treatment with ATO resulted in a notable increase in lipid ROS and MDA levels, which were subsequently reversed by the ferroptosis inhibitors Fer-1 and DFO.

**Data availability statement:** All relevant data are within the manuscript and its Supporting Information files.

**Funding:** This study was supported by Administration of Traditional Chinese Medicine Bureau of Guangdong Province(No.20231403), Basic and Applied Basic Research Foundation of Guangdong Province (No.2022A1515220194), Guangdong University Innovation Team Project (Natural Science 2024KCXTD058), Medical Research Fund of Guangdong Province (No. A2023307, No.B2023280), the scientific research fund of the First People's Hospital of Zhaoqing(No.YJJ-2020-02-03,No.YJJ-2023-02-04,), Zhaoqing Medical College Fund for Young Talent (No.Zqyq22-005, No.Zqyq22-007). The funders had no role in study design, data collection and analysis, decision to publish, or preparation of the manuscript.

**Competing interests:** The authors have declared that no competing interests exist.

Mechanistically, ATO induced ferroptosis by inhibiting GPX4. Furthermore, NRF2 and its downstream targets, HO-1 and SLC7A11, were upregulated during ferroptosis. NRF2 knockdown enhanced lipid peroxidation and ATO-induced cell death.

## Conclusions

ATO significantly promoted ferroptosis in HCC cells, and NRF2 knockdown enhanced the cytotoxic effects of ATO.

---

## Introduction

Primary liver cancer is the sixth most common cancer and fourth leading cause of cancer-related deaths worldwide [1]. It is estimated that more than 1 million individuals will be affected by 2025 [2]. HCC accounts for about 90% of primary liver cancers [3]. As both the incidence and mortality rates of HCC continue to rise annually, liver cancer remains a significant global health challenge. Treatment options for HCC include surgical resection, liver transplantation, radiofrequency ablation, transcatheter arterial chemoembolization(TACE), and systemic therapies. However, the prognosis for HCC remains poor, with a 5-year overall survival rate below 30% [4]. Primary and acquired drug resistance account for the majority of advanced stage-related deaths, and combined with the increasing incidence of HCC, make it a highly lethal disease [5]. Therefore, identifying appropriate therapeutic strategies is important for prevention and treatment of HCC.

Ferroptosis, first identified as an iron-dependent form of regulated cell death, is distinct from apoptosis and necrosis due to its reliance on $Fe^{2+}$-driven reactive oxygen species (ROS) accumulation and lethal lipid peroxidation [6,7]. This unique mechanism enables ferroptosis to selectively eliminate therapy-resistant cancer stem cells and mesenchymal-state tumor cells, offering a promising strategy to overcome chemoresistance in aggressive cancers [8–11]. In HCC, ferroptosis has emerged as a research hotspot. Preclinical studies demonstrate that pharmacologically inducing ferroptosis effectively suppresses HCC proliferation and metastasis [12]. APE1, a key enzyme in DNA repair and redox regulation, enhances ferroptotic cell death and contributes to HCC therapy [13]. Ferroptosis promotes the transformation of M2-to-M1 macrophages, enhancing immunotherapy in HCC [14]. SLC7A11-mediated ferroptosis enhance the efficiency of HCC radiotherapy [15]. Therefore, investigating the mechanisms of ferroptosis in HCC is expected to provide new insights and strategies for the treatment of HCC.

Emerging evidence highlights the therapeutic efficacy of ATO beyond hematologic malignancies, with demonstrated antitumor activity in solid tumors including prostate, breast, gastric, and HCC [16–20]. Mechanistically, ATO suppresses liver cancer stem cell self-renewal and metastatic dissemination by disrupting the SRF/MCM7 axis [20,21].Moreover, a clinical study reported significantly improved median overall survival in patients receiving TACE combined with intravenous ATO [22]. Furthermore, locoregional therapies combined with ATO reduce extrahepatic metastasis rates and

prolong survival in advanced HCC [23]. However, the anticancer mechanism of ATO in HCC requires further investigation. Therefore, elucidating the mechanism of action of ATO could improve its clinical efficacy in the treatment of HCC.

In this study, we found that ATO suppressed HCC progression by triggering ferroptosis via direct targeting of GPX4. Mechanistically, we revealed that NRF2 and its downstream targets, HO-1 and SLC7A11, are upregulated as adaptive resistance mechanisms during ATO treatment. Silencing NRF2 expression with NRF2 small interfering RNA (siRNA) was shown to enhance the antitumor properties of ATO. Our findings identified key effector proteins involved in the ferroptosis pathway induced by ATO, providing insights into a promising therapeutic approach for HCC.

## Materials and methods

### Cell culture

Human hepatocellular carcinoma cell lines HepG2, Huh7, HCCLM3, and MHCC97H were obtained from BeNa Culture Collection (BNCC). The cells were cultured in DMEM (HyClone) supplemented with 10% FBS (Gibco) at 37 °C with 5% $CO_2$. Plasmid or siRNA was transfected into cells using LipoFectMax™ 3000 reagent (ABP Biosciences) respectively.

### Reagents

ATO (5 mg, H20080665) was sourced from Beijing SL Pharmaceutical Co., Ltd. ATO was dissolved in phosphate-buffered saline (PBS). Ferroptosis inhibitors Ferrostatin-1 (Fer-1, HY-100579), deferoxamine (DFO, HY-B1625), and N-acetylcysteine (NAC, HY-B0215) were obtained from MedChemExpress. NRF2-specific siRNA was purchased from RiboBio; the siRNA sequence targeting human NRF2 was 5'-GAGAAAGAATTGCCTGTAA-3'.

### Cell viability assay

HepG2 and Huh7 cells were seeded in 96-well plates at a density of 5,000 cells per well and incubated for 24 h. Various concentrations of ATO were then added to the wells both with and without Fer-1, DFO, and NAC for an additional 24 h. Cell viability was measured using the Cell Counting Kit-8 (CCK-8; Beyotime) according to the manufacturer's instructions.

### Wound healing assay

HepG2 and Huh7 cells were seeded in 12-well culture plates and incubated overnight. Following this, a thin scratch was made using a sterile pipette tip, and the cells were exposed to varying concentrations of ATO. The relative migration widths were then captured and measured using ImageJ software.

### Colony formation assay

HepG2 and Huh7 cells were treated with ATO for 24 hours. Following digestion with trypsin, 1,000 cells were seeded into each well of 6-well plates. The plates were then incubated for approximately 2 weeks until colonies were visibly formed. Subsequently, the colonies were fixed with 4% paraformaldehyde and stained with crystal violet (Beyotime). The number of viable colonies was counted.

### Transwell assay

Transwell chambers (Corning) were positioned in 24-well culture plates previously coated with Matrigel. HepG2 and Huh7 cells were exposed to varying concentrations of ATO. A volume of 200 µL cell suspension in FBS-free DMEM was introduced into the upper Transwell chamber, while 500 µL of DMEM with 10% FBS was added to the lower chambers. Following a 24-hour incubation period, the cells that migrated through the Transwell were fixed with 4% paraformaldehyde, stained with crystal violet, and subsequently visualized and quantified under a microscope.

## Cell cycle assay

HepG2 and Huh7 cells were seeded in 6-well plates and incubated overnight. The cells were trypsinized, fixed in 70% ice-cold ethanol overnight, and then incubated in PBS with RNase and propidium iodide (PI, Sigma) for 30 minutes in the dark. Cell cycle phase analysis was performed using a flow cytometer (Beckman Coulter).

## Iron detection assay

The iron concentration was quantified using a standardized colorimetric assay following the manufacturer's protocol (ab83366, Abcam). Briefly, samples were reacted with the iron probe at 37°C for 60 minutes to allow complete complex formation. The iron concentration was then determined by measuring the absorbance at 593 nm using a microplate reader, with the optical density values being proportional to the iron concentration in the samples.

## Western blot analysis

Following lysis with RIPA buffer (Beyotime), protein extractions underwent separation on SDS-PAGE gels and were subsequently transferred to PVDF membranes(Millipore). The membranes were then blocked with 5% milk for 1 hour at room temperature and immunoblotted with primary antibodies overnight at 4°C. The primary antibodies used were NRF2 (Affility,0639, 1:1000), Tubulin (Proteintech,66240, 1:5000), GPX4 (Proteintech,67763, 1:5000), HO1 (Proteintech,10701, 1:5000),SLC7A11 (Proteintech,26864, 1:2000) and Lamin A/C (Proteintech,10298, 1:1000). Finally, the membranes were incubated with secondary antibodies conjugated to HRP (Sigma,1:5000) and visualized by imaging systems.

## Immunofluorescence

Cells were plated on autoclaved coverslips at an appropriate density and incubated for 24 hours. Subsequently, the cells were treated with PBS and ATO for an additional 24 hours. Following treatment, the cells were fixed with 4% paraformaldehyde (Beyotime) and permeabilized with 0.5% Triton X-100 (Beyotime) for 20 minutes. After blocking with 1% Bovine serum albumin (BSA), the cells were incubated with NRF2 antibody (Affility,0639, 1:1000 dilution) overnight at 4°C, followed by staining with Alexa Fluor 594-conjugated secondary antibodies (DaianA32723, 1:500 dilution) for 1 hour at room temperature in the dark. Finally, the nuclei were stained with DAPI (Abcam, ab104139) for 20 minutes before imaging under a fluorescence microscope.

## Measurement of lipid peroxidation

Cells were plated in six-well culture plates overnight, ATO was added with or without ferroptosis inhibitors for 24 hours. Following this, the cells were incubated with fresh medium containing 5 µM of C11-BODIPY probe (MX5211, MKBio) for 30 minutes at 37°C. Excess probe was removed by washing the cells three times with PBS. The probe-labeled cells were trypsinized and subjected to flow cytometry analysis. Oxidation of the polyunsaturated butadienyl portion of C11-BODIPY resulted in a shift of the fluorescence emission peak from ~590 nm to ~510 nm, proportional to lipid peroxidation generation, and was analyzed using a flow cytometer. C11-BODIPY fluorescence was detected through the FITC channel (510 nm). Mean fluorescence intensity was quantified using FlowJo software.

## Malondialdehyde (MDA) assay

The MDA assay kit (Abcam, ab118970) was utilized to measure the cellular MDA content. Cells were initially seeded into 6-well plates and allowed to culture overnight. Following treatment with ATO and ferroptosis inhibitors for 24 hours. The cells were harvested and lysed for protein quantification. Subsequently, MDA working solution was added to the lysate and heated at 100 °C for 15 minutes. The supernatant was then collected post-centrifugation and measured at 532 nm using a microplate reader.

## Statistical analysis

All experiments were conducted with a minimum of three biological replicates and independently repeated at least three times. The data are presented as mean ± standard deviation (SD) and analyzed using GraphPad Prism 6 software. The values displayed represent the mean and SD (n = 3). An unpaired two-sided Student's t-test and one-way analysis of variance (ANOVA) were used to compare differences between groups. A p-value lower than 0.05 was considered statistically significant.

## Results

### The cytotoxic effect of ATO on HCC

To initially assess the cytotoxic effects of ATO on HCC cell lines, we treated HepG2 cells with concentrations of 0, 1.25, 2.5, 5, 10, 15, 20, 40, and 80 µM, and Huh7 cells with concentrations of 0, 0.75, 1.25, 2.5, 5, 10, 20, and 40 µM. Cell viability was detected using the CCK-8 assay after 24 hours, and the half-maximal inhibitory concentration (IC50) was calculated using GraphPad Prism. To further investigate the cytotoxic effects of ATO on HCC cell lines, both HepG2 and Huh7 cells were exposed to varying concentrations of ATO for 6, 12, 24, 48, and 72 h, with cell viability assessed by the CCK-8 assay. The findings demonstrated a significant decrease in the viability of both HepG2 and Huh7 cells in a dose- and time-dependent manner (Fig 1A–1D). Subsequent experiments utilizing wound healing, colony formation, and Transwell assays revealed that ATO effectively impeded the migration (Fig 1E–1H), colony formation (Fig 1I), and invasion (Fig 1J, 1K) of HCC cells. In conclusion, ATO exhibited notable cytotoxicity and hindered HCC cell progression.

The effect of ATO on the cell cycle was investigated using flow cytometry. S1 Fig A–D illustrate significant changes in the cell cycle of HepG2 and Huh7 cells following ATO treatment, with cells showing arrest at the G1/S-phase compared to the control group. EdU staining was performed to assess the effects of ATO on HCC proliferation at varying concentrations. The proportion of EdU-positive cells was quantified by fluorescence microscopy, revealing a significant decrease in the proportion of EdU-positive cells as the ATO concentration increased (S1 Fig E–H). In summary, ATO effectively inhibited proliferation and induced G1/S-phase arrest in HCC cells.

### ATO-mediated cytotoxicity is dependent on ferroptosis

To determine whether ATO induces ferroptosis in HCC cells, the C11-BODIPY fluorescent probe was used to detect lipid peroxidation levels in HCC cells after ATO treatment. The results are shown in Fig 2A–2D. Significant increases in lipid ROS levels were observed in ATO-treated HCC cells. Furthermore, an MDA assay kit was used to measure MDA levels, the final product of lipid peroxidation, revealing that ATO treatment increased MDA levels in HCC cells (Fig 2E, 2F). To confirm the role of ferroptosis in ATO-induced HCC cell death, ferroptosis inhibitors (DFO, Fer-1) and the antioxidant NAC were combined with ATO to treat HepG2 and Huh7 cells, and lipid peroxidation and MDA levels were detected. Ferroptosis inhibitors partially restored lipid peroxidation levels in HCC cells after ATO treatment (Fig. 2A–2F). Furthermore, iron assay kit was used to detected the $Fe^{2+}$ changes in ATO treated HCC cells, intracellular iron accumulation was observed in ATO-treated HCC cells (S2 Fig). Subsequent CCK-8 assays and morphological analysis revealed that Fer-1, DFO, and NAC partially rescued ATO-induced cell death and restored abnormal cell morphology (Fig 3A–3D).These findings collectively demonstrate that ATO induces ferroptosis in HCC cells by enhancing lipid peroxidation and iron accumulation, and the attenuation of ferroptosis through specific inhibitors treatment partially reverses its cytotoxic effects, underscoring ferroptosis as a pivotal mechanism in ATO-driven HCC cell death.

### NRF2 signaling pathway is activated during ferroptosis

Under specific conditions, NRF2 translocates to the nucleus and activates target genes containing antioxidant response element (ARE) sites [24,25]. Notably, NRF2 orchestrates the activation of numerous cytoprotective genes to suppress

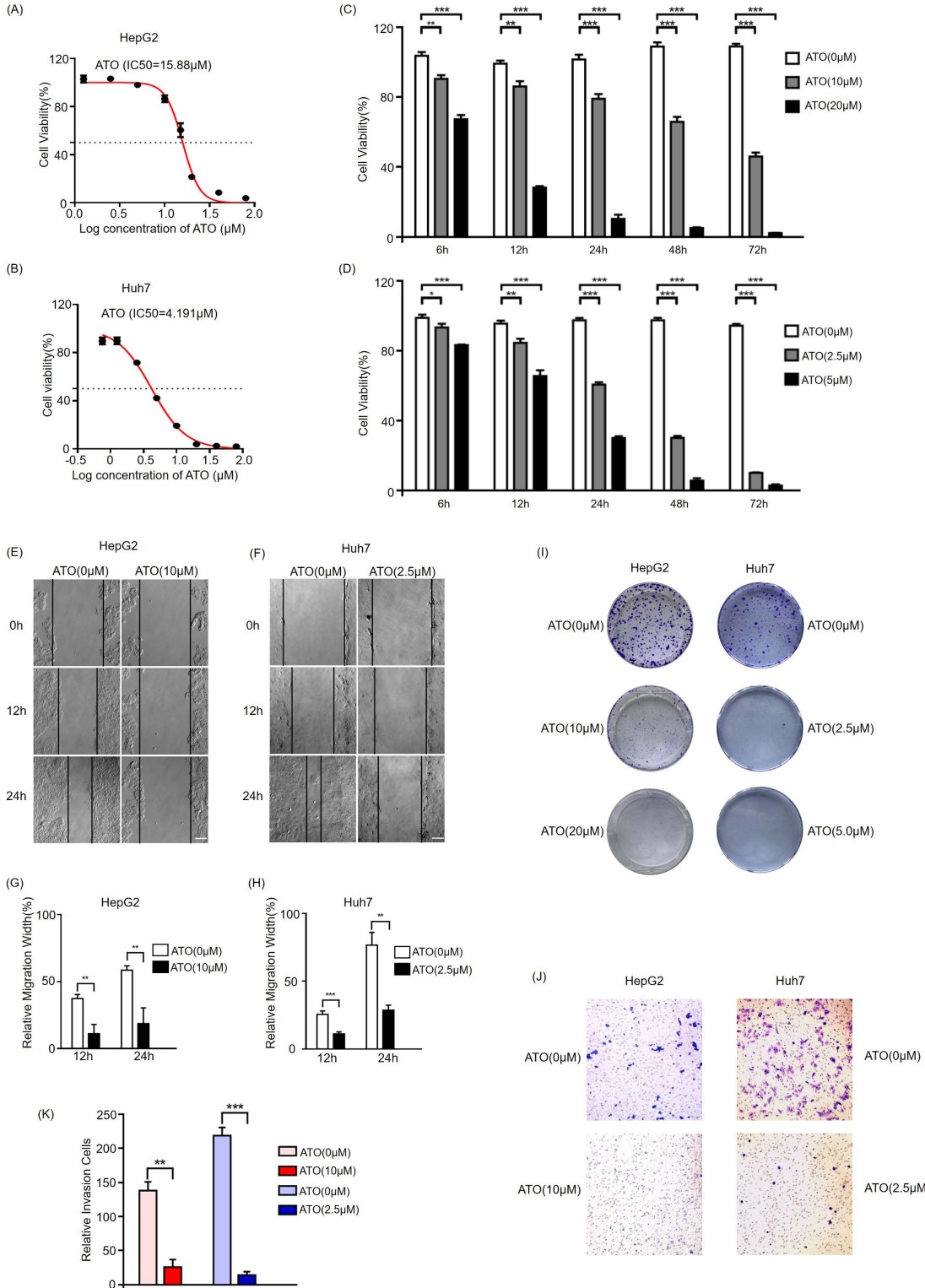

**Fig 1. The cytotoxic effect of ATO on HCCs.** HepG2(A) and Huh7(B) cells were treated with the indicated concentration ATO for 24 hours, then harvested to analyze the cell viability via CCK-8 kit. The half-maximal inhibitory concentration (IC50) was calculated to quantify the 50% inhibitory effect

versus the control. HepG2(C) and Huh7(D) were treated with the indicated concentration ATO for 6,12,24,48, and 72 hours, then harvested to analyze the cell viability via CCK-8 kit. Wound-healing assays were conducted on HepG2(E) and Huh7(F) cells after ATO treatment at 0 h, 12 h, and 24 h, with statistical analysis performed on the relative migration width of the cells. Scale bar = 100μm. Statistical analysis of the relative migration width of Hep-G2(G) and Huh7(H) cells. Representative images of the HepG2 and Huh7 cell colony formation after treatment with the indicated concentration ATO(I). Representative invasive images of transwell assays using HepG2 and Huh7 cells treated with the indicated concentration ATO(J). Bar indicates 50μm. Statistical analysis of the relative invasion cells(K). Error bars indicate SD. * $p < 0.05$, ** $p < 0.01$,*** $p < 0.001$. n = 3.

ferroptosis under stress conditions [26]. Nuclear NRF2 expression was measured by nucleocytoplasmic fractionation. Interestingly, almost all NRF2 was located in the nucleus of HepG2 and Huh7 cells under physiological conditions(Fig 4A–4B). Following ATO treatment, nuclear NRF2 levels increased noticeably (Fig 4C–4D). To investigate the mechanism underlying ATO-induced ferroptosis, effector proteins involved in ferroptosis, such as SLC7A11, HO-1, and GPX4, were analyzed. As illustrated in Fig 4E,4F, the ferroptosis suppressor NRF2 was activated following ATO treatment, resulting in upregulation of key ferroptosis-related genes such as SLC7A11 and HO-1. However, while NRF2 levels increased, GPX4 did not show a corresponding increase. With the increase in doses of ATO, GPX4 levels decreased, suggesting direct regulation by ATO(Fig 4E,4F). Besides, overexpression of GPX4 reduced ferroptosis in ATO-treated HCC cells (S3 Fig). In summary, the induction of ferroptosis in HCC by ATO may be mediated through its direct downregulation of GPX4, while the activated ferroptosis concurrently promotes the upregulation of ferroptosis-resistance genes such as NRF2, HO-1, and SLC7A11.

## NRF2 promotes ATO resistance in HCCs

Studies have demonstrated that NRF2 inhibit the cytotoxic effects of ferroptosis inducers[27,28]. To investigate the relationship between NRF2 expression levels and ATO resistance in HCC cells, NRF2 protein levels were assessed in various HCC cell lines using western blot (Fig 5A,5B), and cell viability following treatment with the same ATO concentration was measured using the CCK-8 assay (Fig 5C). The results revealed that Huh7 cell exhibited the lowest NRF2 expression levels and was the most sensitive to ATO, suggesting a positive correlation between NRF2 expression and ATO resistance in HCC cells. To further validate this finding, HCC cells were transfected with an NRF2 overexpression vector before ATO treatment, and cell viability was subsequently evaluated. It was observed that NRF2 overexpression significantly enhanced the resistance of HCC cells to ATO (Fig 5D–5G).

## Silencing NRF2 increases ATO-triggered ferroptosis

To further validate the protective role of NRF2 in HCC following ATO treatment, HCC cells were transfected with NRF2 siRNA 24 h before ATO treatment, and ferroptosis was subsequently detected. The results demonstrated that after interfering with NRF2 expression, the lipid peroxidation caused by ATO significantly increased (Fig 6A-6D). The Western blotting results showed that after ATO induced ferroptosis in HCC, NRF2 and its downstream ferroptosis-protective genes were upregulated, whereas knockdown of NRF2 led to downregulation of NRF2 and these protective genes. This weakened the protective effect and increased ferroptosis. Although GPX4 is one of the NRF2 downstream ferroptosis-protective genes, its expression is only directly regulated by ATO, with no significant correlation to NRF2. (Fig 6E,6F).

## Silencing NRF2 enhances the sensitivity of HCC cells to ATO

CCK-8 assay showed that the cell viability of siNRF2-transfected cells was significantly reduced after ATO treatment (Fig 7A,7B). Wound healing and colony formation assays further demonstrated that NRF2 downregulation significantly impaired the migratory and colony-forming capacities of ATO treated HCC cells (Fig 7C–7H). This finding underscores the critical role of NRF2 in conferring ATO resistance through its dual functions as an antioxidant and an anti-ferroptotic factor, thereby advancing the understanding of molecular mechanisms regulating ferroptosis in HCC cells.

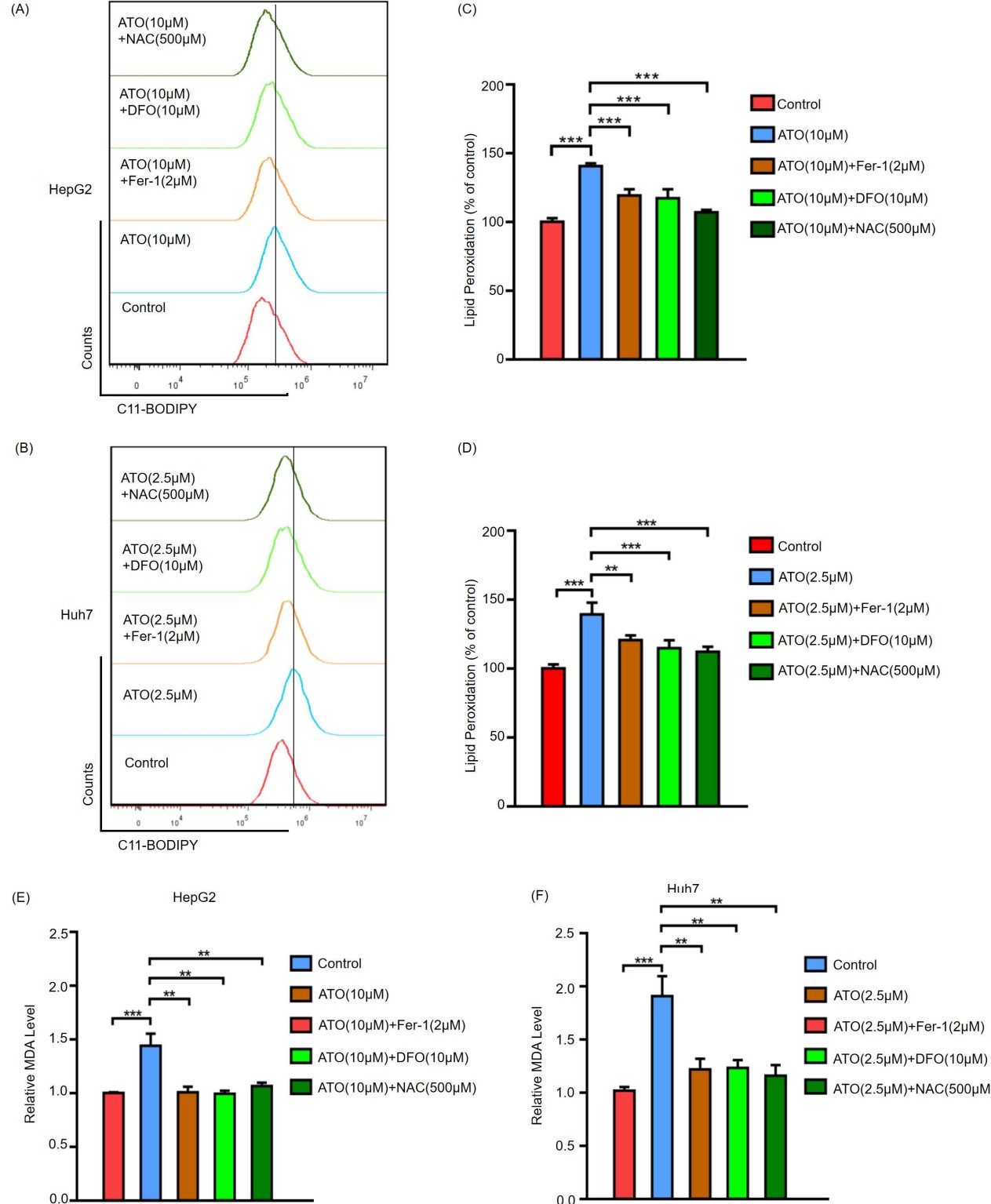

**Fig 2. ATO increases lipid ROS and MDA levels of HCC cells.** HepG2 and Huh7 were treated with the indicated condition for 24 hours, and lipid peroxidation were quantified using C11-BODIPY lipid probe using flow cytometry in HepG2(A) and Huh7(B) cells. Statistical analysis of mean fluorescence intensity for C11-BODIPY lipid probe in HepG2(C) and Huh7(D). HepG2(E) and Huh7(F) were treated with the indicated condition for 24 hours, then harvested to analyze cellular MDA levels. Error bars indicate SD. ** p < 0.01,*** p < 0.001. n = 3.

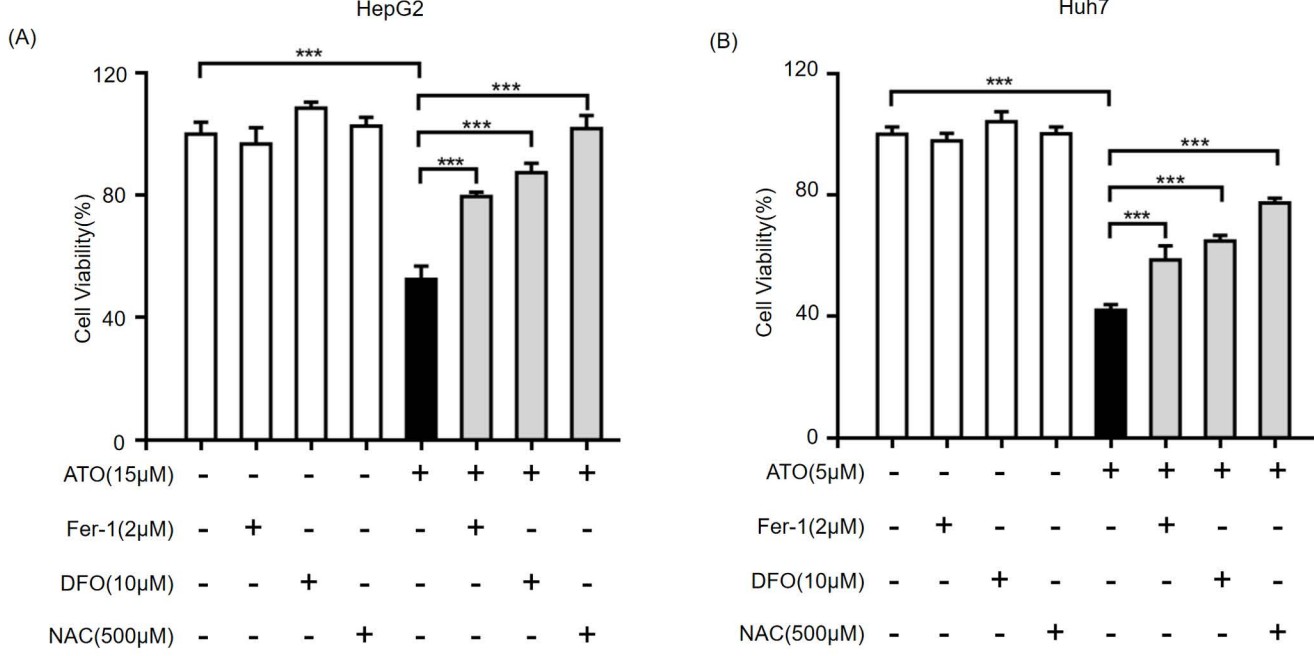

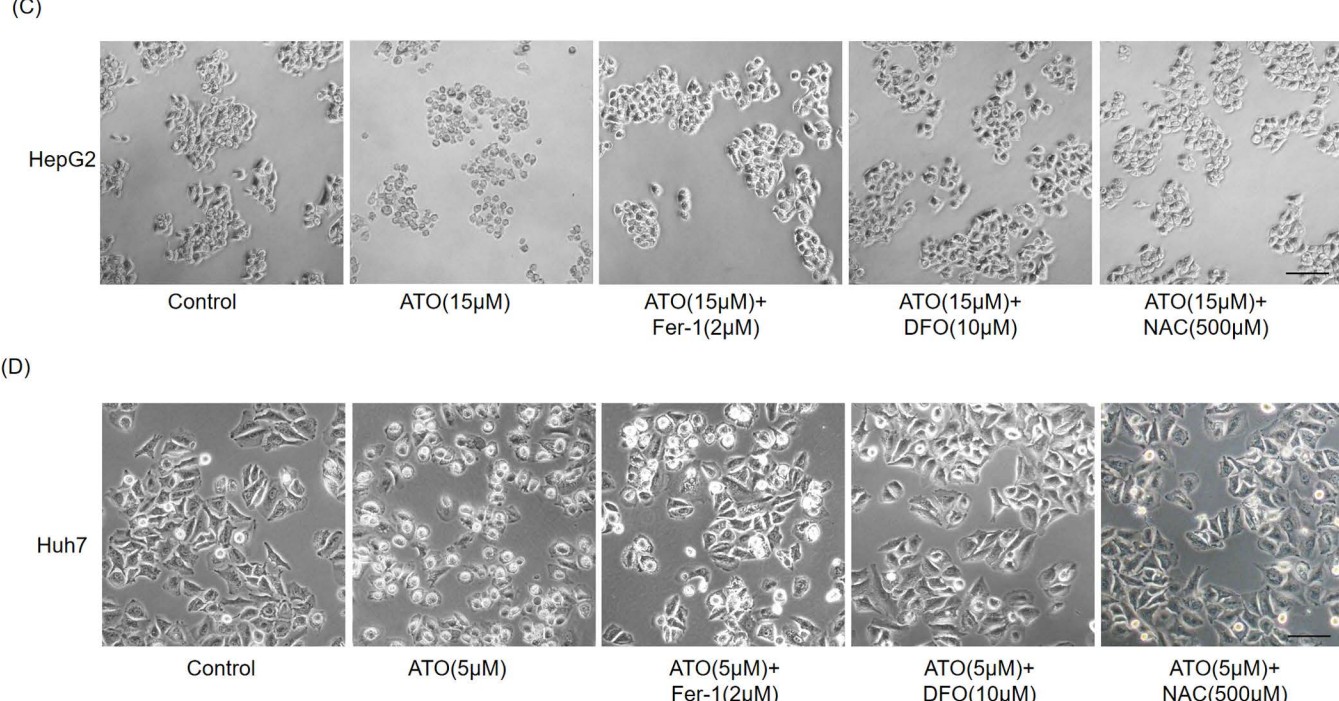

**Fig 3. The cytotoxic effect of ATO on HCCs depend on ferroptosis.** HepG2(A) and Huh7(B) cells were treated with the indicated condition for 24h, then harvested to analyze the cell viability via CCK-8 kit. The HepG2(C) and Huh7(D) cell morphology after treatment with indicated condition for 24h. bar = 100μm. Error bars indicate SD. ***,p < 0.001. n = 3. NC(negative control).

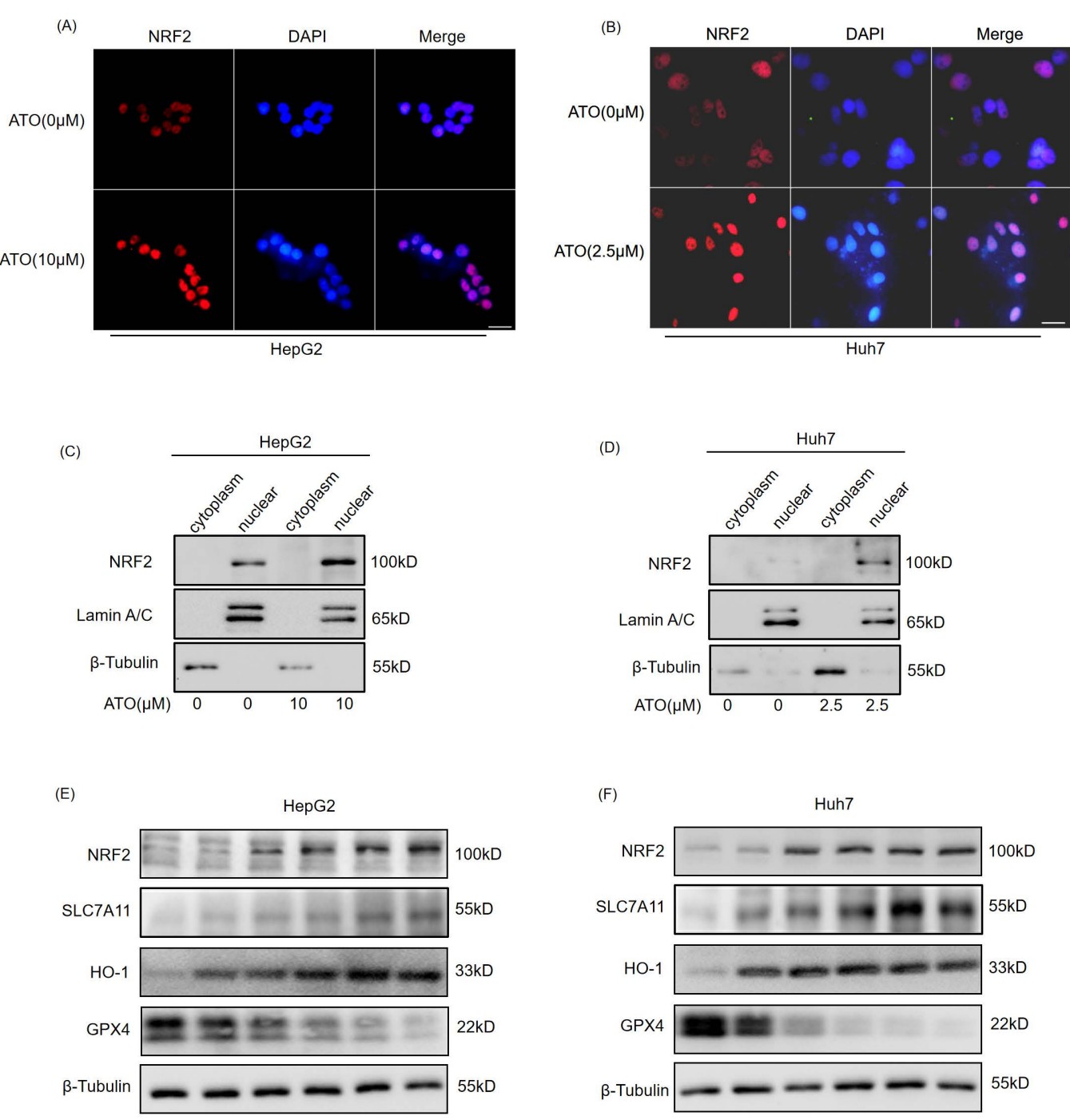

**Fig 4. ATO activates NRF2 in HCC cells.** HepG2(A) and Huh7(B) cells were treated with the indicated concentration ATO for 24h, then harvested to IF experiment with NRF2 antibody. The nuclei were stained with DAPI. Bar indicates 20μm. HepG2(C) and Huh7(D) cells were treated with the indicated concentration ATO for 24h, then harvested for nuclear cytoplasmic separation experiment. Lamin A/C and β-tublin were used as nuclear marker and cytoplasmic marker, respectively. HepG2(E) and Huh7(F) cells were treated with the indicated concentration ATO for 24h, then harvested for western blot.

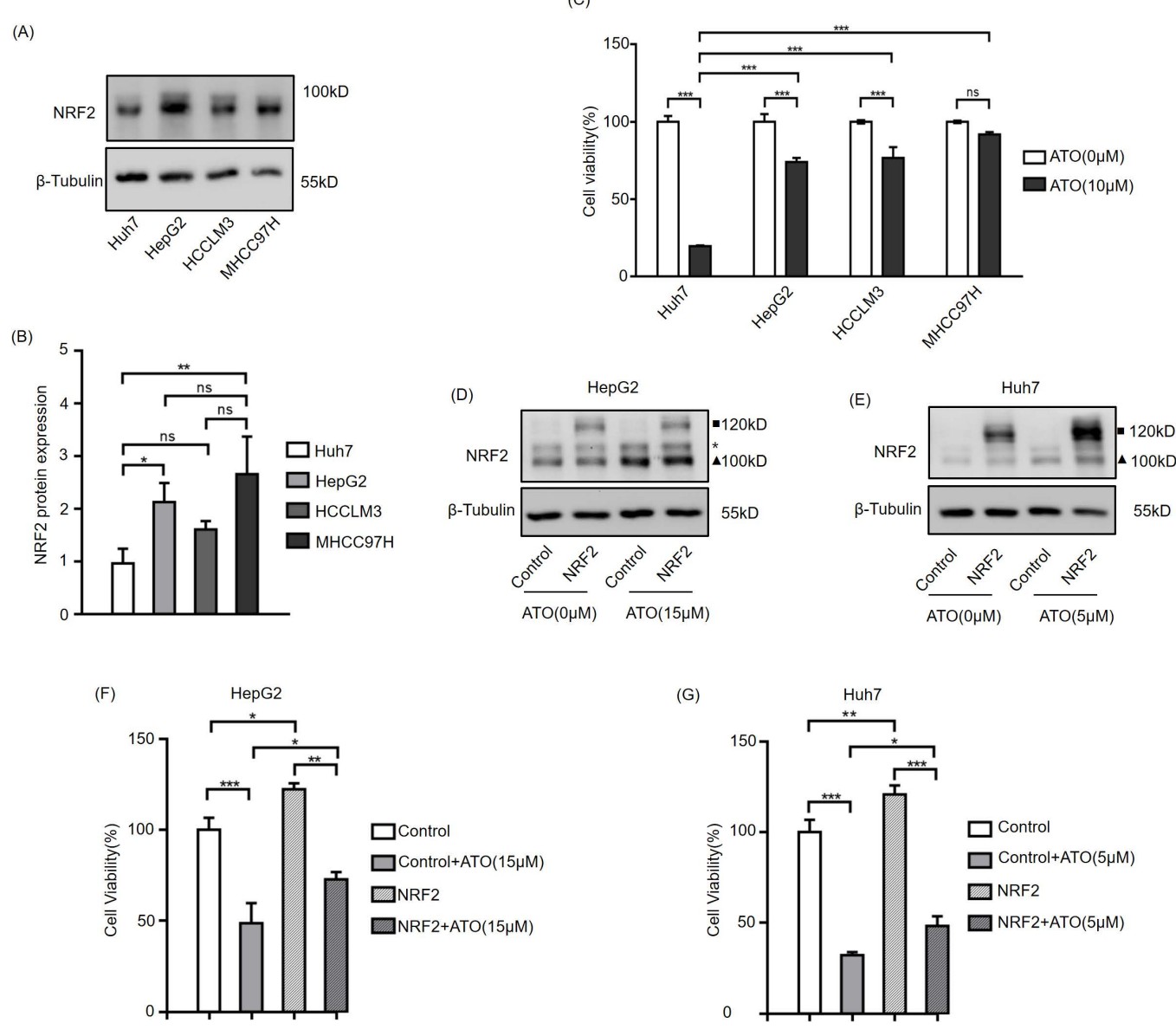

**Fig 5. NRF2 promotes ATO-resistance in HCCs.** HCC cells were harvested to western blot(A). Statistical analysis of the relative NRF2 protein expression of HCC cells(B). HCC cells were treated with ATO (10μM) for 24h, then harvested to CCK-8 assay(C). HepG2(D) and Huh7(E) were transfected with pEGFP-N1-NRF2 or control vectors for 24h, then treated with the indicated concentration ATO for 24h, then harvested to western blot. HepG2(F) and Huh7(G) were transfected with pEGFP-N1-NRF2 or control vectors for 24h, then treated with the indicated concentration ATO for 24h, then harvested to CCK-8 assay. Error bars indicate SD. *,p<0.05, **,p<0.01,***,p<0.001. n=3. ▲ indicates endogenous NRF2, ■ indicates exogenous NRF2, * indicates non-specific bands.

## Discussion

Ferroptosis is a form of cell death characterized by intracellular iron accumulation and lipid peroxidation. The development of ferroptosis inducers is a promising strategy for cancer treatment. This study showed that ATO induces ferroptosis in HCC cells in a time- and dose-dependent manner by enhancing lipid peroxidation levels. The ferroptosis inhibitors

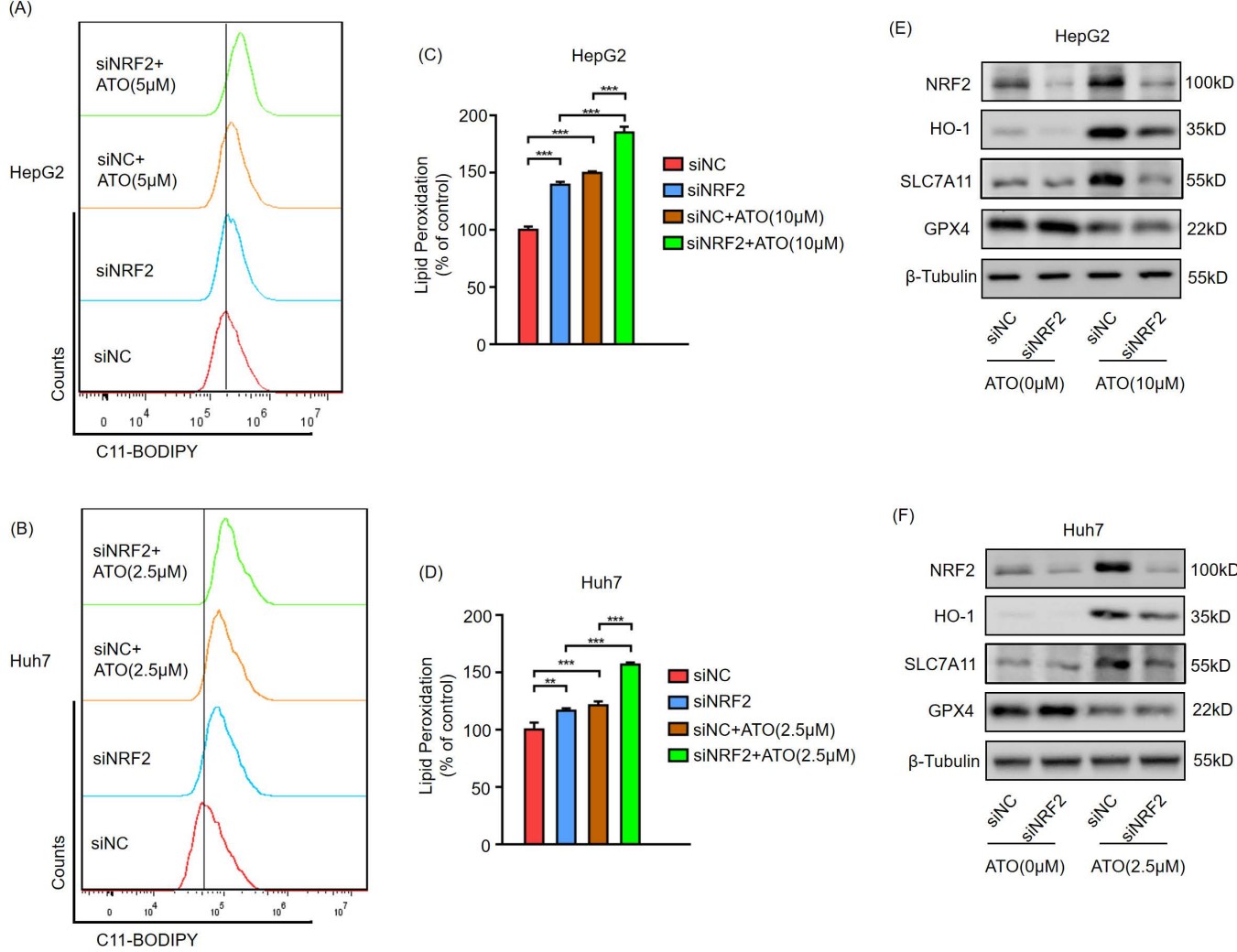

**Fig 6. Silencing NRF2 increases ATO-triggered ferroptosis.** HepG2 (A) and Huh7 (B) cell were transfected with NRF2 siRNA or NC for 24h, then treated with the indicated concentration ATO for 24h, quantified using C11-BODIPY lipid probe. Statistical analysis of the cellular lipid peroxidation in HepG2 (C) and Huh7 **(D)**. HepG2 (E) and Huh7 (F) cell were transfected with NRF2 siRNA or NC for 24h, then treated with the indicated concentration ATO for 24h, then harvested to western blot. Error bars indicate SD. **,p < 0.01,***,p < 0.001. n = 3. NC(negative control).

DFO and Fer-1 partially alleviated this effect (Fig 3A,3B), indicating that other signaling pathways (such as apoptosis) are involved in ATO-induced HCC cell death. Additionally, this study provides new evidence that NRF2 acts as a critical negative regulator of ATO-induced ferroptosis. NRF2 knockdown via siRNA significantly enhanced the anticancer effects of ATO in HCC cells. However, conflicting evidence exists, as several recent studies demonstrate that NRF2 exhibits both tumor-suppressive and tumor-promoting effects [29,30]. Thus, further research is necessary to fully explore this duality, which may help clarify the therapeutic implications of targeting NRF2 in cancer treatment.

NRF2 has a reputation as the master regulator of cellular antioxidant responses, which makes it a key regulator of ferroptosis. Many ferroptosis-related genes, such as SLC7A11, GPX4, and HO-1, are transcriptionally regulated by NRF2 [31–33]. SLC7A11, the subunit of system Xc⁻, is responsible for cystine uptake to synthesize the major ROS scavenger glutathione (GSH) [6]. Consistent with this, SLC7A11 was upregulated by NRF2 activation and downregulated by NRF2

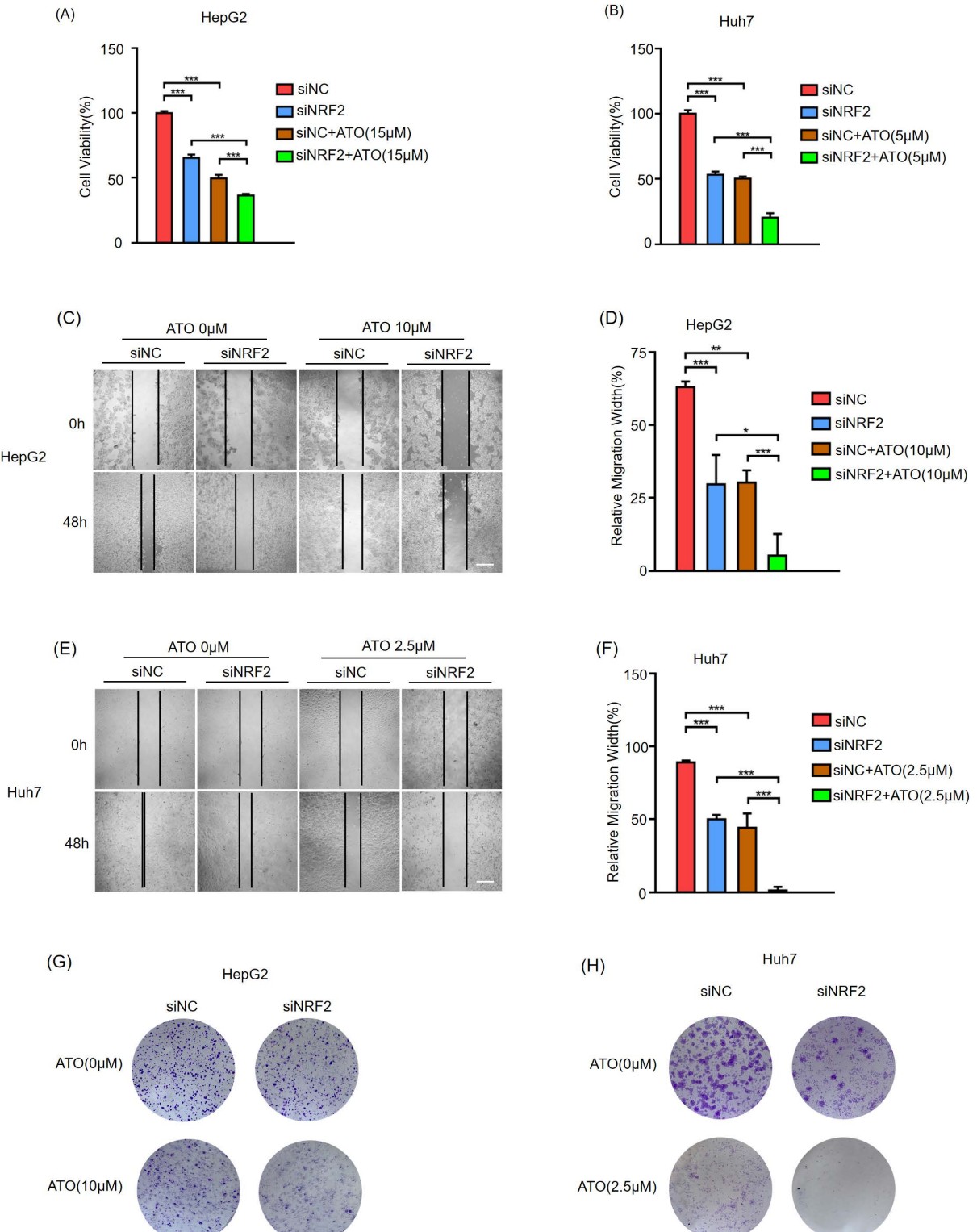

**Fig 7. Silencing NRF2 enhances the sensitivity of HCC cells to ATO.** HepG2(A) and Huh7(B) were transfected with NRF2 siRNA or NC for 24h, and treated with the indicated concentration ATO, then harvested to CCK-8 assay. HepG2(C, D, G) and Huh7(E, F, H) were transfected with NRF2 siRNA

or NC for 24h, and treated with the indicated concentration ATO, then subjected to wound-healing(C, E) and colony formation assays(G, **H**). Scale bar = 100μm. Statistical analysis of the relative migration width of HepG2(D) and Huh7(F) cells. Error bars indicate SD. *,p < 0.05, **,p < 0.01,***,p < 0.001. n = 3. NC(negative control).

knockdown (Figs 4E,4F and 5G,5H). GPX4 catalyzes the reduction of hydroperoxides using GSH as a reductant and plays a central role in ferroptosis by reducing lipid peroxides [34,35]. However, in this study, when NRF2 was upregulated by ATO, GPX4 expression decreased significantly (Figs 4E,4F and 5G,5H), and GPX4 levels remained unchanged following NRF2 knockdown (Figs 4E,4F and 5G,5H). Su et al. reported that ATO induces ferroptosis in neuroblastoma by mediating the transcriptional inhibition of GPX4 [36], suggesting that GPX4 may be directly regulated by ATO rather than NRF2 in HCC. Another NRF2 target, HO-1, serves as a significant intracellular source of iron by detoxifying heme into biliverdin and $Fe^{2+}$ [37]. Although iron overload triggers ferroptosis, biliverdin is further metabolized to the potent antioxidant bilirubin; thus, HO-1 may exert dual roles in ferroptosis depending on the cellular context [38]. The functional significance of elevated HO-1 levels in ATO-treated HCC cells requires further investigation.

ATO administration in solid tumors typically requires higher doses compared to hematologic malignancies. However, such elevated doses may induce side effects like cardiotoxicity. To address this, combining ATO with adjuvant agents or utilizing nanoparticle-based drug delivery systems could enhance therapeutic efficacy while reducing adverse effects. Studies indicate that ATO-loaded biomimetic iron oxide nanoparticles improve HCC treatment outcomes both in vitro and in vivo [39,40]. Additionally, synergistic combinations of ATO with other compounds show promise. Artemisinin attenuates ATO-induced cardiotoxicity [41]. Icariin potentiates ATO cytotoxicity in HCC xenograft models [42]. Crocin mitigates QT prolongation and myocardial damage via the Keap1-Nrf2/HO-1 pathway [43]. Drug delivery strategies also influence therapeutic profiles. For instance, sustained-release formulations of ATO reduce cardiac toxicity [44]. A phase II trial demonstrated that TACE using ATO-loaded microspheres minimizes systemic toxicity through localized drug release and enhanced embolization [45]. Collectively, these findings highlight ATO's potential as a combinatorial therapeutic agent for HCC when integrated with targeted delivery systems or adjuvant therapies.

Systemic therapy has improved the survival rate of patients with HCC, particularly those in advanced stages. However, the high mortality rate of HCC remains a major global public health concern. The primary challenge in therapy is combating drug resistance and preventing recurrence. HCC cells have the ability to undergo a process known as 'redox reset,' resulting in a stronger antioxidant system. ATO induces ROS accumulation and the activation of multiple cell death modalities by different ways. ATO can directly bind to this critical site, disrupting protein function and consequently impairing the regulation of ROS production and clearance[46]. ATO induces structural modification and enzyme inactivation of several oxidative stress biomarker such as catalase, glutathione peroxidases and glutathione reductase,which are considered to be the first line of cellular defence against oxidative damage[47].Additionally, ATO suppresses the synthesis of glutathione (GSH), initiating an enzyme cascade that amplifies ROS generation[48]. Furthermore, ATO elevates ROS levels through the Fenton reaction, which generates highly reactive hydroxyl radicals[49]. Immunofluorescence and nucleocytoplasmic separation experiments showed that nearly all NRF2 protein were located in the nucleus under normal conditions, indicating a high redox balance in HCC cells (Fig 4A-4D). NRF2 plays a key role in the cellular oxidative stress defense system, making it a potential target for disrupting high redox balance and preventing drug resistance in HCC.

However, our research has limitations, and further investigation is needed. All studies were conducted in HCC cell lines; additional in vivo studies using animal models or clinical samples are required to confirm the anticancer effects of ATO in HCC. Furthermore, the mechanism by which ATO regulates GPX4, a key target of NRF2, remains unclear. Additionally, HO-1 and NRF2 levels were significantly altered following ATO treatment, and since both proteins exhibit dual roles in cancer progression, it is crucial to thoroughly investigate their context-dependent functions in HCC treatment. Addressing these limitations would enhance the understanding of ATO's potential as a therapeutic option for HCC.

## Conclusions

Taken together, ATO exhibits anticancer effects against HCC by inducing ferroptosis, potentially through the downregulation of GPX4. Meanwhile, the NRF2-mediated antioxidant pathway is activated during ATO-induced ferroptosis. Critically, our findings demonstrate that inhibition of NRF2 enhances ATO sensitivity in HCC cells. This strategy may facilitate its clinical application in liver cancer therapy.

## Supporting information

**S1 Fig. ATO cause HCC cell cycle arrest and proliferation inhibition.** After ATO treatment for 24 hours, HepG2(A) and Huh7(B) cell cycle were analyzed by flow cytometry. Statistical analysis of the cell cycle count in HepG2(C) and Huh7(D). HepG2(E) and Huh7(F) were treated with indicated concentration ATO for 24 hours, then used for EdU cell proliferation assays,which measure the incorporation of EdU into newly synthesized DNA. The nuclei were stained with DAPI. Bar indicates 50μm. Statistical analysis of relative EdU-positive HepG2(G) and Huh7(H) cells. Error bars indicate SD. **,$p < 0.01$,***,$p < 0.001$. $n = 3$.
(TIFF)

**S2 Fig. ATO enhances intracellular iron ion accumulation.** HepG2(A) and Huh7(B) cells were treated with the indicated conditions for 24h, then harvested to analyze the $Fe^{2+}$ concentration.
(TIFF)

**S3 Fig. GPX4 reduced cell death in ATO-treated HCC cells.** HepG2(A) and Huh7(B) were transfected with pcDNA3.1(-)-GPX4 or control vectors for 24h, then treated with the indicated concentration ATO 24h, then harvested to western blot. HepG2(C) and Huh7(D) were transfected with pcDNA3.1(-)-GPX4 or control vectors for 24h, then treated with the indicated condition for 24h, then harvested to CCK-8 assay. Error bars indicate SD. *,$p < 0.05$, **,$p < 0.01$,***,$p < 0.001$. $n = 3$.
(TIFF)

**S1 Raw images.**
(ZIP)

## Author contributions

**Conceptualization:** Mi Huang, Duanzhuo Li, Zhengzhen Xia, Wenxia Si, Xin Yu, Yi Quan.

**Data curation:** Mi Huang, Duanzhuo Li, Zhengzhen Xia, Shengjie Liao, Minshu Jiang.

**Formal analysis:** Mi Huang, Duanzhuo Li, Wenxia Si, Chao Yuan, Weibin Wu.

**Funding acquisition:** Mi Huang, Duanzhuo Li.

**Investigation:** Zhengzhen Xia.

**Methodology:** Mi Huang, Duanzhuo Li, Zhengzhen Xia, Shengjie Liao, Chao Yuan, Yanli Liao, Weibin Wu.

**Project administration:** Xin Yu, Yi Quan.

**Supervision:** Xin Yu, Yi Quan.

**Validation:** Duanzhuo Li, Zhengzhen Xia, Yanli Liao, Minshu Jiang, Xin Yu, Yi Quan.

**Writing – original draft:** Mi Huang, Duanzhuo Li, Zhengzhen Xia, Shengjie Liao, Wenxia Si, Chao Yuan, Yanli Liao, Weibin Wu, Minshu Jiang, Xin Yu, Yi Quan.

**Writing – review & editing:** Mi Huang, Xin Yu, Yi Quan.

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
