## [Decision Letter · Decision Letter 0]

27 Dec 2024

PONE-D-24-48590Silencing NRF2 Enhances Arsenic Trioxide-Induced Ferroptosis in Hepatocellular Carcinoma CellsPLOS ONE

Dear Dr. Quan,

Thank you for submitting your manuscript to PLOS ONE. After careful consideration, we feel that it has merit but does not fully meet PLOS ONE’s publication criteria as it currently stands. Therefore, we invite you to submit a revised version of the manuscript that addresses the points raised during the review process.

We believe that the manuscript lacks clarity and precision in several aspects, and the presented data require further completion and strengthening. We invite you to submit a revised version of the manuscript that thoroughly addresses all the issues raised during the review process (please refer to the comments below from the Reviewer and Editor).

 Please submit your revised manuscript by Feb 10 2025 11:59PM. If you will need more time than this to complete your revisions, please reply to this message or contact the journal office at plosone@plos.org . Please include the following items when submitting your revised manuscript:

We look forward to receiving your revised manuscript.

Kind regards,

Prof. Pierre Bobé

Academic Editor

PLOS ONE

Journal Requirements:

This study was supported by Administration of Traditional Chinese Medicine Bureau of Guangdong Province(No.20231403), Basic and Applied Basic Research Foundation of Guangdong Province (No.2022A1515220194), Guangdong University Innovation Team Project (Natural Science 2024KCXTD058), Medical Research Fund of Guangdong Province (No. A2023307, No.B2023280), the scientific research fund of the First People's Hospital of Zhaoqing(No.YJJ-2020-02-03,No.YJJ-2023-02-04,), Zhaoqing Medical College Fund for Young Talent (No.Zqyq22-005, No.Zqyq22-007). 

Additional Editor Comments:

1) Ferroptosis is one of the many mechanisms by which arsenic trioxide induces cellular stress in both normal and cancer cells. The authors should discuss in their manuscript the various ways in which ATO contributes to the production of elevated levels of reactive oxygen species (ROS), rather than limiting their discussion to ferroptosis alone. For example, the ability of arsenic trioxide to form covalent bonds with specific proteins, thereby altering their structure and function, has been extensively studied as a key factor in ROS production in ATO-treated cells.

2) Have the authors investigated whether arsenic trioxide affects mechanisms beyond ferroptosis? If so, what were the results?

3) The authors reported an IC50 of approximately 16 µM for HepG2 cells and 10 µM for Huh7 cells when evaluating the efficacy of arsenic trioxide (ATO) in targeting hepatocellular carcinoma cells. This raises a critical question about the suitability of these concentrations for therapeutic use of ATO in solid tumors. It is noteworthy that the standard dose of ATO administered for acute promyelocytic leukemia is roughly ten to twenty times lower. The significant disparity in effective concentrations suggests that while ATO may show effectiveness against HCC cells in vitro, such high concentrations may not be practical for clinical applications. This concern is heightened even when considering combinations with other agents to lower the dosage, due to the risk of adverse effects, including QT interval prolongation, linked to ATO therapy. This pharmacological issue underscores the need for a more comprehensive discussion regarding the therapeutic dosing of ATO, particularly in the context of its use for solid tumors.

Reviewers' comments:

Reviewer's Responses to Questions

**Comments to the Author**

1. Is the manuscript technically sound, and do the data support the conclusions?

Reviewer #1: Partly

2. Has the statistical analysis been performed appropriately and rigorously? 

Reviewer #1: Yes

3. Have the authors made all data underlying the findings in their manuscript fully available?

Reviewer #1: Yes

4. Is the manuscript presented in an intelligible fashion and written in standard English?

Reviewer #1: No

5. Review Comments to the Author

Reviewer #1: Huang et al explores the mechanism underlying ATO sensitivity in hepatocellular carcinoma. The authors described how ATO induces ferroptosis by inhibiting GPX4 and increased downstream targets Nrf2, HO-1, and SLC7A11. Molecular inhibition of Nrf2 further increased of ferroptosis in HCC cells by inhibiting SLC7A11 and HO-1 downstream targets. ATO induces ferroptosis in HCC and Inhibiting NRF2 increase ATO sensitivity in HCC are already known. The novelty of this study is ATO induces ferroptosis in HCC by inhibiting GPX4 and NRF2 inhibition increases ATO mediated ferroptosis in HCC.

Major concerns

Interpretation of the figures and data was not evident throughout the results section.

1. The Bodipy flow plots (fig2A, 2b and Fig 5A, 5C) converted into percentage lipid peroxidation (Fig2c, 2d and fig5b, 5d). Please calculate the mean fluorescence intensity for the Bodipy flow plots and show the significance

2. Typically, Nrf2 expression in cancer cells is associated with chemoresistance. Do you observe resistance to ATO in these cell lines when Nrf2 is expressed? Have you observed a difference in the expression of Nrf2 in ATO-resistant and sensitive cell lines?

3. The Nrf2 immunofluorescence image in Fig. 4a without ATO treatment appears dull (with decreased brightness) in both cell lines. Quantify the Nrf2 expression with and without ATO treatment and plot the difference.

4. Explain why multiple bands are observed in Fig. 4e Western blot of the HepG2 cell line treated with ATO, whereas the previous blots in Fig. 4c and Fig. 4d for Nrf2 show a prominent single band.

5. Studies have shown that increased SLC7A11 expression enhances GPX4 activity, which inhibits lipid peroxidation-mediated ROS. Consequently, ferroptosis is suppressed, leading to chemoresistance. In this manuscript, Fig. 5E and 5F showed that ATO treatment increases SLC7A11 transporter expression, while Fig. 5A and 5C demonstrate that ferroptosis is also increased. How will explain this?

6. Fig. 5g: SLC7A11 Western blot appears unclear/hazy.

7. Fig 5G and Fig 5H – Nrf2, HO-1, GPX4 and SLC7A11 blots looks identical for both the cells lines HepG2 and Huh7

8. Targeted GPX4 overexpression could be carried out. If GPX4 overexpression decreases ferroptosis after ATO treatment, this proves the role of GPX4 in this context.

9. Extending the findings to animal models will be helpful.

Minor comments:

1. To rewrite the sentence – line 29-32 (not clear)

2. Ferroptosis in HCC is a well-established area. The introduction part doesn’t talk about this. Needs to be updated.

3. What is the specificity of ferroptosis in this context? How do you distinguish this from other form of cell death like apoptosis, necrosis or autophagy?

4. How ATO treatment after NRF2 inhibition affects colony formation capacity and wound healing?

5. Do ferroptosis inducers in combination with ATO increase cell death (with and without NRF2 inhibition)?

6. Was intracellular iron load quantified?

7. What is the significance of ferroptosis mediated cell death in the context of ATO resistant HCC cells?

6. PLOS authors have the option to publish the peer review history of their article (what does this mean? ). If published, this will include your full peer review and any attached files.

**Do you want your identity to be public for this peer review?** For information about this choice, including consent withdrawal, please see our Privacy Policy .

Reviewer #1: No

---

## [Author Response · Author response to Decision Letter 1]

22 Feb 2025

Dear Editors and Reviewers,

Thank you for your letter and for the reviewers’ comments concerning our manuscript entitled “Silencing NRF2 Enhances Arsenic trioxide-Induced Ferroptosis in Hepatocellular Carcinoma Cells(PONE-D-24-48590)”. Those comments are all valuable and very helpful for revising and improving our paper, as well as the important guiding significance to our researches. We fully understand the need to strengthen our discussion and experimental sections. We are currently working on adding additional experiments to provide further evidence to support our findings. Additionally,we will expand our discussion to address the limitations of our current work and potential directions for future research.

All changes have been highlighted in the revised manuscript using blue color for your convenience. We believe these revisions have significantly improved the quality and clarity of the manuscript. Please let us know if there are any additional adjustments or clarifications needed. We look forward to your feedback and hope that the revised version meets the journal’s standards for publication. Thank you once again for your support and guidance throughout this process.

Yours Sincerely,

Yi Quan, Ph.D.

Department of Oncology, Zhaoqing First People’s Hospital,No.9 Donggang East Road, Guangdong,People’s Republic of China

E-mial: quany_i@sina.com

Tel: +86 18902368663;

Fax: 0758-2823506;

According to the reviewer’s comments and recommendation, we have done the corresponding revisions in the manuscript. The point-to-point responses are as follow:

Editor Comments:

Comment 1:Ferroptosis is one of the many mechanisms by which arsenic trioxide induces cellular stress in both normal and cancer cells. The authors should discuss in their manuscript the various ways in which ATO contributes to the production of elevated levels of reactive oxygen species (ROS), rather than limiting their discussion to ferroptosis alone. For example, the ability of arsenic trioxide to form covalent bonds with specific proteins, thereby altering their structure and function, has been extensively studied as a key factor in ROS production in ATO-treated cells.

Response:The authors appreciate the reviewer for the critical analysis and constructive comments. This comment is indeed fascinating and warrants further investigation.We have already add the various ways in which ATO contributes to the production of elevated levels of ROS in the discussion section.

Cysteine thiol serves as the functional site in most redox proteins. ATO can directly bind to this critical site, disrupting protein function and consequently impairing the regulation of ROS production and clearance (PMID: 12471082).ATO induces structural modification and enzyme inactivation of several oxidative stress biomarker such as catalase, glutathione peroxidases and glutathione reductase,which considered to be the first line of cellular defence against oxidative damage (PMID: 29498595).Additionally, ATO suppresses the synthesis of glutathione (GSH), initiating an enzyme cascade that amplifies ROS generation (PMID: 12124315). Furthermore, ATO elevates ROS levels through the Fenton reaction, which generates highly reactive hydroxyl radicals (PMID: 17970581). Recent studies have also revealed that ATO promotes ROS production by enhancing calpain-mediated degradation of transglutaminase 2, a key protein involved in cancer cell survival (PMID: 37446117).

Comment 2:Have the authors investigated whether arsenic trioxide affects mechanisms beyond ferroptosis? If so, what were the results?

Response:Thank you for your advice. We detected the cell viability induced by ATO with VAD (Z-VAD-FMK, apoptosis inhibitor) ,Nec-1 (necrostatin-1, necroptosis inhibitor),3-MA (3-Methyladenine, autophagy inhibitor) and Fer-1(Ferrostatin-1, ferroptosis inhibitor) respectively. As shown in figure below, ATO-induced cell death was attenuated by VAD and Fer-1,also Moderated by 3-MA slightly. Indeed, ATO induces cell death through multiple pathways, particularly apoptosis and ferroptosis. Given that the apoptosis pathway has been extensively studied (PMID: 31997633,PMID: 32920515), we chose ferroptosis as the focus of our research.

Comment 3: The authors reported an IC50 of approximately 16 µM for HepG2 cells and 10 µM for Huh7 cells when evaluating the efficacy of arsenic trioxide (ATO) in targeting hepatocellular carcinoma cells. This raises a critical question about the suitability of these concentrations for therapeutic use of ATO in solid tumors. It is noteworthy that the standard dose of ATO administered for acute promyelocytic leukemia is roughly ten to twenty times lower. The significant disparity in effective concentrations suggests that while ATO may show effectiveness against HCC cells in vitro, such high concentrations may not be practical for clinical applications. This concern is heightened even when considering combinations with other agents to lower the dosage, due to the risk of adverse effects, including QT interval prolongation, linked to ATO therapy. This pharmacological issue underscores the need for a more comprehensive discussion regarding the therapeutic dosing of ATO, particularly in the context of its use for solid tumors.

Response:The authors appreciate the reviewer for the critical analysis.

The doses of ATO is indeed much higher in solid tumor than acute promyelocytic leukemia. ATO has a narrow window of therapeutic opportunity in respect of side-effects. In recent years, various new therapeutic strategies have been investigated to enhance the therapeutic efficacy of ATO and reduce its toxicity.

1)ATO as a therapeutic agent or a chemosensitizer in HCC patients prolonged survival. ATO intravenous infusion combined with transcatheter arterial chemoembolization (TACE) effectively controlled pulmonary metastasis and prolonged overall survival in patients with HCC.Locoregional therapy combined with ATO prevents extrahepatic metastasis and prolongs the survival time for primary HCC patients,and the combination strategy was safe and effective in treatment (PMID:27517972, PMID:25504506,PMID:26033499).

2)Different drug delivery methods can affect drug concentrations(PMID: 16154855 ) . The slow delivery of ATO can alleviate cardiac side effects.A phase 2 trial announced that TACE using microsphere beads loaded with ATO can reduce its adverse events due to the more complete embolization and more sustained drug release(PMID: 32096197)

3)Combinaion of ATO and other agents has considerable therapeutic efficacy in HCC.Artemisinin could attenuate the cardiotoxicity induced by ATO(PMID: 30179145).Icariin treatment potentiated the cytotoxicity of ATO in xenograft mice model.(PMID: 23975599). Crocin not only ameliorated QT prolongation but also improved myocardial damage via the Keap1-Nrf2/HO-1 pathway(PMID: 32920515).

4)Application of nanoformulations. Through encapsulation in different biocompatible nanoparticle platforms, ATO demonstrated enhanced therapeutic efficacy against cancer while exhibiting reduced toxic side effects.(PMID: 35033089,PMID: 37162268). ATO-loaded magnetic nanoparticles with HCC cell membranes exhibited a desirable ferroptosis-based strategy for safe and reliable HCC therapeutics(PMID: 36695492)

These findings highlight the potential of combination therapies to enhance the therapeutic benefits of ATO while mitigating its adverse effects.

Reviewer #1

Major concerns

1.The Bodipy flow plots (fig2A, 2b and Fig 5A, 5C) converted into percentage lipid peroxidation (Fig2c, 2d and fig5b, 5d). Please calculate the mean fluorescence intensity for the Bodipy flow plots and show the significance

Response:We apologize for the lack of clarity in the Methods section. We conducted the statistical analysis exactly as suggested by the reviewer. We have now added a detailed explanation in the Methods section to address this.

2. Typically, Nrf2 expression in cancer cells is associated with chemoresistance. Do you observe resistance to ATO in these cell lines when Nrf2 is expressed? Have you observed a difference in the expression of Nrf2 in ATO-resistant and sensitive cell lines?

Response:Thanks for the kindly advice. To investigate the relationship between NRF2 expression levels and ATO resistance in HCC cells, NRF2 protein levels were assessed in various HCC cell lines using western blot (Figure 5A,B), and cell viability following treatment with the same ATO concentration was measured using the CCK8 assay (Figure 5C). The results revealed that Huh7 cell exhibited the lowest NRF2 expression levels and was the most sensitive to ATO, suggesting a positive correlation between NRF2 expression and ATO resistance in HCC cells.

2.The Nrf2 immunofluorescence image in Fig. 4a without ATO treatment appears dull (with decreased brightness) in both cell lines. Quantify the Nrf2 expression with and without ATO treatment and plot the difference.

Response:Thank you. This is our negligence. We have adjusted the fluorescence intensity. The Nrf2 expression have already quantified according to the western blot with and without ATO treatment,as shown in figure below.

4. Explain why multiple bands are observed in Fig. 4e Western blot of the HepG2 cell line treated with ATO, whereas the previous blots in Fig. 4c and Fig. 4d for Nrf2 show a prominent single band.

Response:thanks for the comment.The multiple bands were non-specific bands.The non-specific bands for the NRF2 antibody exist in all blot. For the reason of different exposure time, the non-specific bands was stronger in Figure 4E, weaker in Figure 4C and Figure 4D.

5.Studies have shown that increased SLC7A11 expression enhances GPX4 activity, which inhibits lipid peroxidation-mediated ROS. Consequently, ferroptosis is suppressed, leading to chemoresistance. In this manuscript, Fig. 5E and 5F showed that ATO treatment increases SLC7A11 transporter expression, while Fig. 5A and 5C demonstrate that ferroptosis is also increased. How will explain this?

Response:Thanks for your question.Under normal conditions, when drugs or other proteins act on SLC7A11, SLC7A11 upregulates GPX4, and the upregulated GPX4 exerts its inhibitory effect on ferroptosis, thereby preventing lipid peroxidation. However, the situation here is different. Our experimental results demonstrate that ATO directly acts on GPX4, leading to its downregulation. Even if the upstream regulators of GPX4, such as NRF2 and SLC7A11, are upregulated, they cannot reverse the downregulation of GPX4. Consequently, the downregulation of GPX4 directly triggers the occurrence of lipid peroxidation.

6. Fig. 5g: SLC7A11 Western blot appears unclear/hazy.

Response:Thank you. According to the comments, we have repeated the results and replaced all SLC7A11 western blot.

7. Fig 5G and Fig 5H – Nrf2, HO-1, GPX4 and SLC7A11 blots looks identical for both the cells lines HepG2 and Huh7

Response:Thank you. All the experiments were carried out in HepG2 and Huh7 respectively. They have the same trend of change blots in Fig 5G and Fig 5H, maybe the blots looks like identical, but the images are different.

8. Targeted GPX4 overexpression could be carried out. If GPX4 overexpression decreases ferroptosis after ATO treatment, this proves the role of GPX4 in this context.

Response:Thank you. We transfected pcDNA3.1(-)-GPX4 plasmid into HepG2 and Huh7, and detected cell viability after ATO treatment. As shown in Figure S3, GPX4 overexpression indeed decreases ferroptosis.

9.Extending the findings to animal models will be helpful.

Response:The authors appreciate the reviewer for the valuable comments and constructive comments.It will be meaningful confirm the result in vivo animal models. However, model animal ethical approval and experiments may takes longer time. Besides,adding the complexity of model organisms would require additional financial and temporal resources that are beyond our current allocation. All the limitations have been mentioned in the discussion part.We are enthusiastic about the potential of extending our research to include clinical samples and model organisms in future studies. We plan to pursue these avenues once we have established a robust foundation through our in vitro work and have secured additional funding and collaborative opportunities.

Minor comments:

1. To rewrite the sentence – line 29-32 (not clear)

Response:We are sorry for the unclear description.We have already rewrite the sentences in revision.

2. Ferroptosis in HCC is a well-established area. The introduction part doesn’t talk about this. Needs to be updated.

Response:We sincerely appreciate the valuable comments. Ferroptosis in HCC is updated in the introduction part in the revision.

3. What is the specificity of ferroptosis in this context? How do you distinguish this from other form of cell death like apoptosis, necrosis or autophagy?

Response: Thank you.

We also detected the cell viability induced by ATO with VAD (Z-VAD-FMK, apoptosis inhibitor), Nec-1 (necrostatin-1, necroptosis inhibitor), 3-MA (3-Methyladenine, autophagy inhibitor) and Fer-1(Ferrostatin-1, ferroptosis inhibitor) respectively. As shown in figure below, ATO-induced cell death was attenuated by VAD and Fer-1,also Moderated by 3-MA slightly. Indeed, ATO induces cell death through multiple pathways, particularly apoptosis and ferroptosis. Given that the apoptosis pathway has been extensively studied (PMID: 31997633,PMID: 32920515), we chose ferroptosis as the focus of our research.

4. How ATO treatment after NRF2 inhibition affects colony formation capacity and wound healing?

Response:Thank you. According to the comments, experiments were conducted. The data of colony formation capacity and wound healing after NRF2 inhibition is shown in Figure7C-H.

5. Do ferroptosis inducers in combination with ATO increase cell death (with and without NRF2 inhibition)?

Response:Thanks. We selected erastin as the ferroptosis inducer for CCK8 assay, and the data are presented in the figure below. The combination of ferroptosis inducer with ATO can increase HCC cell death; however, when ferroptosis inducer added to ATO-treated HCC cells with NRF2 already inhibited, although an increase in HCC cell death is observed, it is not significant.

6. Was intracellular iron load quantified?

Response: Thank you for the comment. Intracellular iron load was quantified by iron detected kit as shown in Figure S2.

7. What is the significance of ferroptosis mediated cell death in the context of ATO resistant HCC cells?

Response:Thank you.

Inducing cell death via the ferroptosis pathway represents a promising strategy for overcoming resistance to ATO in HCC. Our findings reveal that in ATO-resistant HCC, a protective mechanism against ferroptosis is activated, as evidenced by the upregulation of the NRF2 pathway. Upon targeted interference with NRF2, HCC cells lose this protective capacity and become significantly more susceptible to ATO. This enhanced sensitivity not only allows for the use of lower ATO dosages but also minimizes the drug's associated side effects, thereby improving therapeutic efficacy while reducing toxicity.

---

## [Decision Letter · Decision Letter 1]

18 Mar 2025

PONE-D-24-48590R1Silencing NRF2 Enhances Arsenic Trioxide-Induced Ferroptosis in Hepatocellular Carcinoma CellsPLOS ONE

Dear Dr. Quan,

Thank you for submitting your manuscript to PLOS ONE. After careful consideration, we feel that it has merit but does not fully meet PLOS ONE’s publication criteria as it currently stands. Therefore, we invite you to submit a revised version of the manuscript that addresses the points raised during the review process.

We believe that the authors have partially addressed the reviewer's concerns (see the reviewer's comments below).

We look forward to receiving your revised manuscript.

Kind regards,

Prof. Pierre Bobé

Academic Editor

PLOS ONE

Reviewers' comments:

Reviewer's Responses to Questions

**Comments to the Author**

1. If the authors have adequately addressed your comments raised in a previous round of review and you feel that this manuscript is now acceptable for publication, you may indicate that here to bypass the “Comments to the Author” section, enter your conflict of interest statement in the “Confidential to Editor” section, and submit your "Accept" recommendation.

Reviewer #1: (No Response)

2. Is the manuscript technically sound, and do the data support the conclusions?

Reviewer #1: Partly

3. Has the statistical analysis been performed appropriately and rigorously? 

Reviewer #1: Yes

4. Have the authors made all data underlying the findings in their manuscript fully available?

Reviewer #1: Yes

5. Is the manuscript presented in an intelligible fashion and written in standard English?

Reviewer #1: Yes

6. Review Comments to the Author

Reviewer #1: The Bodipy flow plots (Fig. 2A, 2B, and Fig. 5A, 5C) have been converted into percentage lipid peroxidation (Fig. 2C, 2D, and Fig. 5B, 5D). Please calculate the mean fluorescence intensity for the Bodipy flow plots and show the statistical significance.- this is not addressed satisfactorily

Fig. 5G and Fig. 5H – The Nrf2, HO-1, GPX4, and SLC7A11 blots appear identical for both cell lines, HepG2 and Huh7.- the response to this comment is not convincing

7. PLOS authors have the option to publish the peer review history of their article (what does this mean? ). If published, this will include your full peer review and any attached files.

**Do you want your identity to be public for this peer review?** For information about this choice, including consent withdrawal, please see our Privacy Policy .

Reviewer #1: No

---

## [Author Response · Author response to Decision Letter 2]

21 Mar 2025

The point-to-point responses are as follow:

Reviewer #1: The Bodipy flow plots (Fig. 2A, 2B, and Fig. 5A, 5C) have been converted into percentage lipid peroxidation (Fig. 2C, 2D, and Fig. 5B, 5D). Please calculate the mean fluorescence intensity for the Bodipy flow plots and show the statistical significance.- this is not addressed satisfactorily

Response

Thanks. We apologize for any confusion caused by the unclear description in the previous response. The lipid peroxidation sensor C11-BODIPY was used according to the manuscript. Oxidation of the polyunsaturated butadienyl portion of C11-BODIPY resulted in a shift of the fluorescence emission peak from ~590 nm to ~510 nm, proportional to lipid peroxidation generation, and was analyzed using a flow cytometer. C11-BODIPY fluorescence was detected through the FITC channel (510 nm). Mean fluorescence intensity was quantified using FlowJo software, and bar graphs in Figures 2C-D and 5B-D were generated based on the mean fluorescence intensity values from each experimental group. In these graphical representations, the mean fluorescence intensity values directly correspond to the lipid peroxidation levels of individual samples, serving as a quantitative measure of cellular membrane oxidation status across experimental conditions.

Fig. 5G and Fig. 5H – The Nrf2, HO-1, GPX4, and SLC7A11 blots appear identical for both cell lines, HepG2 and Huh7.- the response to this comment is not convincing

Response

Thanks. We apologize for the misunderstanding in our previous response and for the inaccurate explanation provided. Despite the data originating from different samples of two cell lines, the blots look identical. To address potential misinterpretation, we have replaced the Western blot image with repeated experimental data. The results are presented in Fig. 6 of the revised manuscript.

Additionally, All the original figures of the western blots have been submitted to the journal. The full uncropped original Western blot images for this figure are provided.

---

## [Decision Letter · Decision Letter 2]

27 Mar 2025

Silencing NRF2 Enhances Arsenic Trioxide-Induced Ferroptosis in Hepatocellular Carcinoma Cells

PONE-D-24-48590R2

Dear Dr. Quan,

We’re pleased to inform you that your manuscript has been judged scientifically suitable for publication and will be formally accepted for publication once it meets all outstanding technical requirements.

Kind regards,

Prof. Pierre Bobé

Academic Editor

PLOS ONE

Additional Editor Comments (optional):

Reviewers' comments:

Reviewer's Responses to Questions

**Comments to the Author**

1. If the authors have adequately addressed your comments raised in a previous round of review and you feel that this manuscript is now acceptable for publication, you may indicate that here to bypass the “Comments to the Author” section, enter your conflict of interest statement in the “Confidential to Editor” section, and submit your "Accept" recommendation.

Reviewer #1: All comments have been addressed

2. Is the manuscript technically sound, and do the data support the conclusions?

Reviewer #1: Yes

3. Has the statistical analysis been performed appropriately and rigorously? 

Reviewer #1: Yes

4. Have the authors made all data underlying the findings in their manuscript fully available?

Reviewer #1: Yes

5. Is the manuscript presented in an intelligible fashion and written in standard English?

Reviewer #1: Yes

6. Review Comments to the Author

Reviewer #1: (No Response)

7. PLOS authors have the option to publish the peer review history of their article (what does this mean? ). If published, this will include your full peer review and any attached files.

**Do you want your identity to be public for this peer review?** For information about this choice, including consent withdrawal, please see our Privacy Policy .

Reviewer #1: No

---

## [Editor Report · Acceptance letter]

PONE-D-24-48590R2

PLOS ONE

Dear Dr. Quan,

I'm pleased to inform you that your manuscript has been deemed suitable for publication in PLOS ONE. Congratulations! Your manuscript is now being handed over to our production team.

Kind regards,

on behalf of

Prof Pierre Bobé

Academic Editor

PLOS ONE